# Characterisation of 22445 patients attending UK emergency departments with suspected COVID-19 infection: Observational cohort study

Steve Goodacre[1]*, Ben Thomas[1], Ellen Lee[1], Laura Sutton[1], Amanda Loban[1], Simon Waterhouse[1], Richard Simmonds[1], Katie Biggs[1], Carl Marincowitz[1], Jose Schutter[1], Sarah Connelly[1], Elena Sheldon[1], Jamie Hall[1], Emma Young[1], Andrew Bentley[2], Kirsty Challen[3], Chris Fitzsimmons[4], Tim Harris[5], Fiona Lecky[1], Andrew Lee[1], Ian Maconochie[6], Darren Walter[7]

1 School of Health and Related Research (ScHARR), University of Sheffield, Sheffield, United Kingdom, 2 Intensive Care, Manchester University NHS Foundation Trust, Wythenshawe Hospital, Manchester, United Kingdom, 3 Emergency Department, Lancashire Teaching Hospitals NHS Foundation Trust, Preston, United Kingdom, 4 Emergency Department, Sheffield Children's NHS Foundation Trust, Sheffield, United Kingdom, 5 Emergency Department, Barts Health NHS Trust, London, United Kingdom, 6 Emergency Department, Imperial College Healthcare NHS Trust, London, United Kingdom, 7 Emergency Department, Manchester University NHS Foundation Trust, Wythenshawe Hospital, Manchester, United Kingdom

* s.goodacre@sheffield.ac.uk

**Data Availability Statement:** Data are available at DOI: 10.15131/shef.data.13194845.

## Abstract

### Background

Hospital emergency departments play a crucial role in the initial assessment and management of suspected COVID-19 infection. This needs to be guided by studies of people presenting with suspected COVID-19, including those admitted and discharged, and those who do not ultimately have COVID-19 confirmed. We aimed to characterise patients attending emergency departments with suspected COVID-19, including subgroups based on sex, ethnicity and COVID-19 test results.

### Methods and findings

We undertook a mixed prospective and retrospective observational cohort study in 70 emergency departments across the United Kingdom (UK). We collected presenting data from 22445 people attending with suspected COVID-19 between 26 March 2020 and 28 May 2020. Outcomes were admission to hospital, COVID-19 result, organ support (respiratory, cardiovascular or renal), and death, by record review at 30 days. Mean age was 58.4 years, 11200 (50.4%) were female and 11034 (49.6%) male. Adults (age >16 years) were acutely unwell (median NEWS2 score of 4), frequently had limited performance status (46.9%) and had high rates of admission (67.1%), COVID-19 positivity (31.2%), organ support (9.8%) and death (15.5%). Children had much lower rates of admission (27.4%), COVID-19 positivity (1.2%), organ support (1.4%) and death (0.3%). Similar numbers of men and women presented to the ED, but men were more likely to be admitted (72.9% v 61.4%), require organ

**Funding:** Steve Goodacre received funding from the United Kingdom National Institute for Health Research Health Technology Assessment (HTA) programme (project reference 11/46/07, https://www.nihr.ac.uk/explore-nihr/funding-programmes/health-technology-assessment.htm). The funder played no role in the study design; in the collection, analysis, and interpretation of data; in the writing of the report; and in the decision to submit the article for publication. The views expressed are those of the authors and not necessarily those of the NHS, the NIHR or the Department of Health and Social Care.

**Competing interests:** All authors declare grant funding to their employing institutions from the National Institute for Health Research (NIHR), as outlined under financial disclosure information. SG is Deputy Director of the NIHR Health Technology Assessment (HTA) Programme, which funded the study, and chairs the NIHR HTA commissioning committee. These competing interests do not alter our adherence to PLOS ONE policies on sharing data and materials.

support (12.2% v 7.7%) and die (18.2% v 13.0%). Black or Asian adults tended to be younger than White adults (median age 54, 50 and 67 years), were less likely to have impaired performance status (43.1%, 26.8% and 51.6%), be admitted to hospital (60.8%, 57.3%, 69.6%) or die (11.6%, 11.2%, 16.4%), but were more likely to require organ support (15.9%, 14.3%, 8.9%) or have a positive COVID-19 test (40.8%, 42.1%, 30.0%). Adults admitted with suspected and confirmed COVID-19 had similar age, performance status and comorbidities (except chronic lung disease) to those who did not have COVID-19 confirmed, but were much more likely to need organ support (22.2% v 8.9%) or die (32.1% v 15.5%).

## Conclusions

Important differences exist between patient groups presenting to the emergency department with suspected COVID-19. Adults and children differ markedly and require different approaches to emergency triage. Admission and adverse outcome rates among adults suggest that policies to avoid unnecessary ED attendance achieved their aim. Subsequent COVID-19 confirmation confers a worse prognosis and greater need for organ support.

## Registration

ISRCTN registry, ISRCTN56149622, http://www.isrctn.com/ISRCTN28342533.

## Introduction

Hospital emergency departments (ED) have played a crucial role during the COVID-19 pandemic in receiving acutely ill patients, determining the need for admission and critical care, and providing emergency treatment. International [1, 2] and national [3–6] guidelines have been developed for the emergency management of suspected COVID-19.

Studies of hospitalised cases with COVID-19 [7–10] inform the emergency management of suspected COVID-19 but have important limitations. First, patients typically present with suspected rather than proven COVID-19. This presentation includes many patients with characteristics of COVID-19, who need urgent care, but do not ultimately have the virus. Second, emergency management involves differentiating those with severe illness who require hospital admission from those with mild or moderate illness who can be managed at home. Appropriate management of this heterogeneous population is an important challenge that needs to be informed by relevant data.

The Pandemic Respiratory Infection Emergency System Triage (PRIEST) study collected data from consecutive patients attending EDs across the UK with suspected COVID-19. We aimed to characterise patients attending EDs with suspected COVID-19, including subgroups based on sex, ethnicity and COVID-19 results.

## Materials and methods

The PRIEST study was originally set up and piloted as the Pandemic Influenza Triage in the Emergency Department (PAINTED) study as part of the UK National Institute for Health Research (NIHR) pandemic portfolio of studies to be activated in the event of an influenza pandemic [11, 12]. It was developed into the PRIEST study and expanded to include other respiratory infections in response to the emerging COVID-19 pandemic.

We undertook an observational cohort study of adults and children attending the ED with suspected COVID-19 infection. Patients were included if the assessing clinician recorded that the patient had suspected COVID-19 in the ED records or completed a standardised assessment form for suspected COVID-19 patients. The clinical diagnostic criteria for COVID-19 during the study were of fever ($\geq$ 37.8˚C) and at least one of the following respiratory symptoms, which must be of acute onset: persistent cough (with or without sputum), hoarseness, nasal discharge or congestion, shortness of breath, sore throat, wheezing, sneezing. We did not seek consent to collect data but information about the study was provided in the ED and patients could withdraw their data at their request. Patients with multiple presentations to hospital were only included once, using data from the first presentation identified by research staff.

Baseline characteristics at presentation to the ED were recorded prospectively, using a standardised assessment form developed and piloted for the PAINTED study [12] that doubled as a clinical record (SF_S1 Appendix: Standardised Data Collection Form), or retrospectively, through research staff extracting data onto the standardised form using the clinical records. Research staff collected follow-up data onto a standardised follow-up form (SDF_S2 Appendix: Follow-up Form) using clinical records up to 30 days after presentation. They then entered data onto a secure online database managed by the Sheffield Clinical Trials Research Unit (CTRU).

Patients who died or required respiratory, cardiovascular or renal support were classified as having an adverse outcome. Patients who survived to 30 days without requiring respiratory, cardiovascular or renal support were classified as having no adverse outcome. Respiratory support was defined as any intervention to protect the patient's airway or assist their ventilation, including non-invasive ventilation or acute administration of continuous positive airway pressure. It did not include supplemental oxygen alone or nebulised bronchodilators. Cardiovascular support was defined as any intervention to maintain organ perfusion, such as inotropic drugs, or invasively monitor cardiovascular status, such as central venous pressure or pulmonary artery pressure monitoring, or arterial blood pressure monitoring. It did not include peripheral intravenous cannulation or fluid administration. Renal support was defined as any intervention to assist renal function, such as haemofiltration, haemodialysis or peritoneal dialysis. It did not include intravenous fluid administration.

The sample size was determined by the size and severity of the pandemic, but was originally planned to involve recruiting 20,000 patients across 40 sites. This was expected to include 200 with an adverse outcome, based on a 1% prevalence of adverse outcome in a previous study undertaken during the 2009 H1N1 pandemic.

This paper presents a descriptive analysis of the cohort. We calculated a National Early Warning Score (2nd version, NEWS2) for adults, to provide an overall assessment of acute illness severity on a scale from zero to 20, based on respiratory rate, oxygen saturation, systolic blood pressure, heart rate, level of consciousness and temperature [13]. We calculated a modified Paediatric Observation Priority Score (POPS) for children for the same purpose, with a scale from zero to 14, based on respiratory rate, oxygen saturation, heart rate, level of consciousness, temperature, breathing and past medical history (excluding the gut feeling parameter) [14]. We undertook descriptive analysis of subgroups based on age, sex and ethnicity. We also compared the characteristics and outcomes of admitted patients with positive COVID-19 testing to those with negative or no testing.

## Ethical approval

The North West—Haydock Research Ethics Committee gave a favourable opinion on the PAINTED study on 25 June 2012 (reference 12/NW/0303) and on the updated PRIEST study

on 23rd March 2020. The Confidentiality Advisory Group of the Health Research Authority granted approval to collect data without patient consent in line with Section 251 of the National Health Service Act 2006.

### Patient and public involvement

The Sheffield Emergency Care Forum (SECF) is a public representative group interested in emergency care research [15]. Members of SECF advised on the development of the PRIEST study and two members joined the Study Steering Committee. Patients were not involved in the recruitment to and conduct of the study. We are unable to disseminate the findings to study participants directly.

### Results

The PRIEST study recruited 22484 patients from 70 EDs across 53 sites between 26 March 2020 and 28 May 2020. We included 22445 in the analysis after excluding 39 who requested withdrawal of their data. The mean age was 58.4 years, 11200 (50.4%) were female, 11034 (49.6%) male (211 missing), and ethnicity was 15198 (84.7%) UK/Irish/other white, 1150 (6.4%) Asian, 692 (3.9%) Black/African/Caribbean, 328 (1.8%) mixed/multiple ethnic groups, 570 (3.2%) other ethnic groups and 4507 unknown (missing data or preferring not to say). After ED assessment COVID-19 was considered the most likely diagnosis for 14400 (67.2% of those with non-missing data). Fig 1 shows that hourly presentations between 11:00 and 18:00 were around four times the night-time rate.

Table 1 shows the baseline characteristics, presenting features and physiology of adults and children in the cohort, and Table 2 shows the admission decisions and adverse outcomes for adults and children.

Adults with suspected COVID-19 were acutely unwell, with a lower IQR oxygen saturation of 94% and an upper IQR respiratory rate of 26/minute, and had high rates of admission (67.1%), organ support (9.8%) and death (15.5%). Children with suspected COVID-19 also

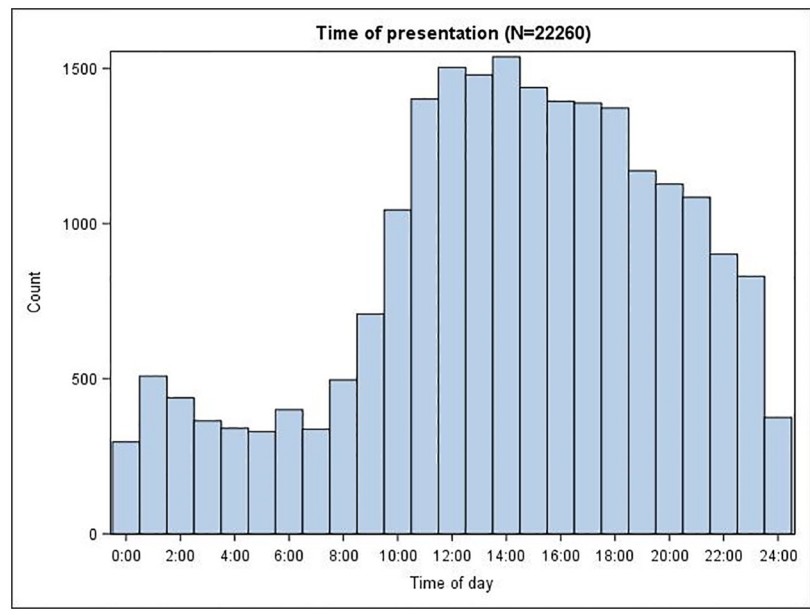

**Fig 1. Time of presentation to the ED.**

**Table 1. Baseline characteristics, presenting features and physiology of adults (N = 20908) and children (N = 1530)†.**

| Characteristic | Statistic/level | Adults | Children |
|---|---|---|---|
| Age (years) | N | 20908 | 1530 |
| | Mean (SD) | 62.4 (19.7) | 3.6 (4.2) |
| | Median (IQR) | 64 (48,79) | 2 (0,6) |
| Sex | Missing | 193 | 18 |
| | Male | 10209 (49.3%) | 821 (54.3%) |
| | Female | 10506 (50.7%) | 691 (45.7%) |
| Ethnicity | Missing/prefer not to say | 4215 | 290 |
| | UK/Irish/other white | 14243 (85.3%) | 950 (76.6%) |
| | Asian | 1044 (6.3%) | 106 (8.5%) |
| | Black/African/Caribbean | 640 (3.8%) | 52 (4.2%) |
| | Mixed/multiple ethnic groups | 247 (1.5%) | 81 (6.5%) |
| | Other | 519 (3.1%) | 51 (4.1%) |
| Presenting features | Cough | 12994 (62.1%) | 580 (37.9%) |
| | Shortness of breath | 15586 (74.5%) | 314 (20.5%) |
| | Fever | 10282 (49.2%) | 1222 (79.9%) |
| Symptom duration (days) | N | 18890 | 1442 |
| | Mean (SD) | 7.9 (8.9) | 4.3 (5.9) |
| | Median (IQR) | 5 (2,10) | 2 (1,5) |
| Heart rate (beats/min) | N | 20477 | 1482 |
| | Mean (SD) | 94.9 (21.6) | 137.2 (28.4) |
| | Median (IQR) | 93 (80,108) | 138 (118,157) |
| Respiratory rate (breaths/min) | N | 20363 | 1473 |
| | Mean (SD) | 23.3 (7) | 33.1 (10.3) |
| | Median (IQR) | 22 (18,26) | 32 (26,40) |
| Systolic BP (mmHg) | N | 20315 | 376 |
| | Mean (SD) | 134.6 (24.9) | 107.9 (15.2) |
| | Median (IQR) | 133 (118,149) | 109 (98,117) |
| Diastolic BP (mmHg) | N | 20228 | 366 |
| | Mean (SD) | 78.2 (16.1) | 65.3 (12.4) |
| | Median (IQR) | 78 (68,88) | 64 (58,73) |
| Temperature (°C) | N | 20248 | 1485 |
| | Mean (SD) | 37.1 (1.1) | 37.5 (1.1) |
| | Median (IQR) | 37 (36.4,37.8) | 37.4 (36.7,38.3) |
| Oxygen saturation (%) | N | 20649 | 1498 |
| | Mean (SD) | 94.7 (6.8) | 97.7 (3.1) |
| | Median (IQR) | 96 (94,98) | 98 (97,99) |
| Glasgow Coma Scale | N | 15434 | 506 |
| | Mean (SD) | 14.6 (1.4) | 14.9 (0.9) |
| | Median (IQR) | 15 (15,15) | 15 (15,15) |
| AVPU | Missing | 2391 | 120 |
| | Alert | 17580 (94.9%) | 1394 (98.9%) |
| | Verbal | 640 (3.5%) | 11 (0.8%) |
| | Pain | 183 (1%) | 3 (0.2%) |
| | Unresponsive | 114 (0.6%) | 2 (0.1%) |

†N = 7 omitted due to missing age

**Table 2. Outcomes of adults (N = 20908) and children (N = 1530).**

| Outcome | Level | Adult N (%) | Child N (%) |
|---|---|---|---|
| Admitted at initial assessment | Missing | 45 | 3 |
| | No | 6866 (32.9%) | 1109 (72.6%) |
| | Yes | 13997 (67.1%) | 418 (27.4%) |
| Respiratory pathogen | COVID-19 | 6521 (31.2%) | 19 (1.2%) |
| | Influenza | 27 (0.1%) | 2 (0.1%) |
| | Other | 1721 (8.2%) | 237 (15.5%) |
| | None identified | 12639 (60.5%) | 1272 (83.1%) |
| Mortality status | Missing | 20 | 3 |
| | Alive | 17642 (84.5%) | 1523 (99.7%) |
| | Dead | 3246 (15.5%) | 4 (0.3%) |
| | Death with organ support* | 693 (21.3%) | 0 (0%) |
| | Death with no organ support* | 2553 (78.7%) | 4 (100%) |
| Organ support | Respiratory | 1944 (9.3%) | 18 (1.2%) |
| | Cardiovascular | 517 (2.5%) | 8 (0.5%) |
| | Renal | 218 (1%) | 2 (0.1%) |
| | Any | 2058 (9.8%) | 22 (1.4%) |

*Denominator = total deaths in category

presented with abnormal physiology, but had low rates of admission, organ support and mortality. Adults tended to present with cough and breathlessness, while children tended to present with fever. Very few children had a positive test for COVID-19, compared with almost a third of adults.

Fig 2 shows the NEWS2 score for adults and Fig 3 shows the POPS score for children. The median (inter-quartile range [IQR]) NEWS2 score was 4 (2, 7) for adults and the median POPS score was 1 (1, 3) for children.

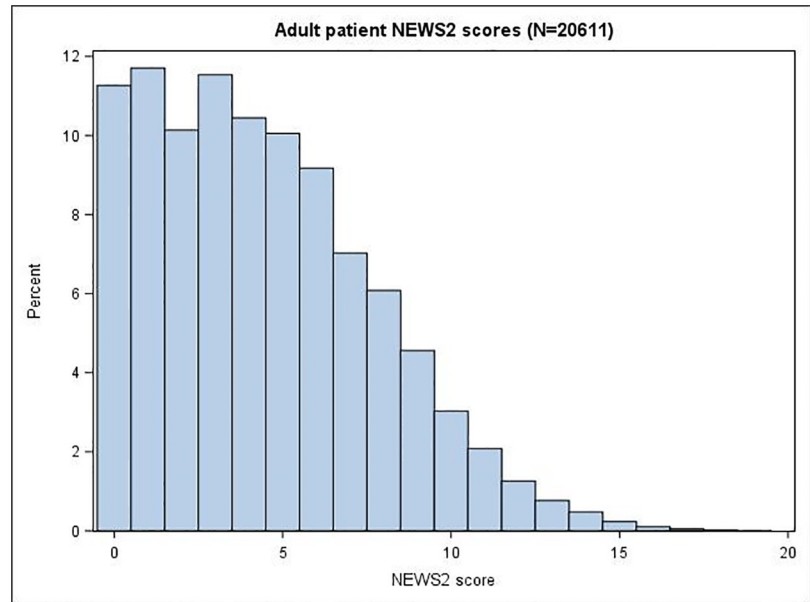

**Fig 2. Adult patients NEWS2 scores.**

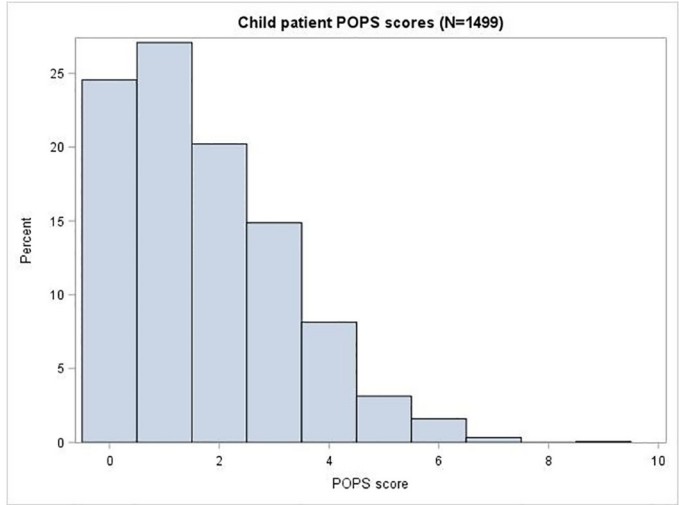

**Fig 3. Child patient POPS scores.**

Table 3 shows that adults with suspected COVID-19 had substantial co-morbidities (30.8% with hypertension and 19.7% with diabetes) and almost half were recorded as having some limitation of normal activities. A substantial proportion (19.3%) had a Do Not Attempt Resuscitation decision recorded on or before the day of presentation.

Table 4 shows that men tended to be older than women, have slightly more severe illness, and were more likely to have hypertension, heart disease, diabetes or chronic lung disease, while women were more likely to have asthma. Men and women attended the ED in similar numbers, but men were more likely to be admitted, have positive COVID-19 testing, require organ support and die.

**Table 3. Co-morbidities, performance status and Do Not Attempt Resuscitation decisions for adults (N = 20908).**

| Characteristic | Level | N (%) |
|---|---|---|
| Comorbidities | Hypertension | 6437 (30.8%) |
| | Heart Disease | 4702 (22.5%) |
| | Diabetes | 4129 (19.7%) |
| | Other chronic lung disease | 3767 (18%) |
| | Asthma | 3410 (16.3%) |
| | Renal impairment | 1934 (9.3%) |
| | Active malignancy | 1120 (5.4%) |
| | Immunosuppression | 631 (3%) |
| | Steroid therapy | 557 (2.7%) |
| | No Chronic disease | 5798 (27.7%) |
| Performance status | Missing | 1080 |
| | Unrestricted normal activity | 10541 (53.2%) |
| | Limited strenuous activity, can do light | 2373 (12%) |
| | Limited activity, can self care | 2781 (14%) |
| | Limited self care | 2649 (13.4%) |
| | Bed/chair bound, no self care | 1484 (7.5%) |
| DNAR in place after ED assessment | | 4029 (19.3%) |

**Table 4. Characteristics and outcomes of male (N = 10209) and female (N = 10506) adults†.**

| Characteristic | Statistic/level | Adult men | Adult women |
|---|---|---|---|
| Age (years) | N | 10209 | 10506 |
| | Mean (SD) | 64 (18.3) | 60.8 (20.9) |
| | Median (IQR) | 66 (51,79) | 61 (45,79) |
| Presenting features | Cough | 6406 (62.7%) | 6473 (61.6%) |
| | Shortness of breath | 7646 (74.9%) | 7811 (74.3%) |
| | Fever | 5224 (51.2%) | 4969 (47.3%) |
| Symptom duration (days) | N | 9216 | 9501 |
| | Mean (SD) | 7.6 (8.5) | 8.3 (9.2) |
| | Median (IQR) | 5 (2,10) | 5 (2,10) |
| Respiratory rate (breaths/min) | N | 9951 | 10228 |
| | Mean (SD) | 23.7 (7.3) | 22.8 (6.7) |
| | Median (IQR) | 22 (18,27) | 21 (18,26) |
| Oxygen saturation (%) | N | 10094 | 10367 |
| | Mean (SD) | 94.2 (7) | 95.1 (6.6) |
| | Median (IQR) | 96 (93,98) | 97 (94,98) |
| NEWS2 score | N | 10118 | 10304 |
| | Mean (SD) | 4.7 (3.4) | 4.1 (3.2) |
| | Median (IQR) | 4 (2,7) | 4 (1,6) |
| Comorbidities | Hypertension | 3356 (32.9%) | 3013 (28.7%) |
| | Heart Disease | 2718 (26.6%) | 1945 (18.5%) |
| | Diabetes | 2343 (23%) | 1747 (16.6%) |
| | Other chronic lung disease | 1981 (19.4%) | 1760 (16.8%) |
| | Asthma | 1261 (12.4%) | 2117 (20.2%) |
| | Renal impairment | 1029 (10.1%) | 888 (8.5%) |
| | Active malignancy | 659 (6.5%) | 453 (4.3%) |
| | Immunosuppression | 294 (2.9%) | 333 (3.2%) |
| | Steroid therapy | 248 (2.4%) | 305 (2.9%) |
| | No Chronic disease | 2659 (26%) | 3080 (29.3%) |
| Performance status | Missing | 530 | 539 |
| | Unrestricted normal activity | 5005 (51.7%) | 5437 (54.6%) |
| | Limited strenuous activity, can do light | 1216 (12.6%) | 1134 (11.4%) |
| | Limited activity, can self care | 1420 (14.7%) | 1339 (13.4%) |
| | Limited self care | 1315 (13.6%) | 1308 (13.1%) |
| | Bed/chair bound, no self care | 723 (7.5%) | 749 (7.5%) |
| Admitted at initial assessment | Missing | 22 | 23 |
| | No | 2765 (27.1%) | 4043 (38.6%) |
| | Yes | 7422 (72.9%) | 6440 (61.4%) |
| Respiratory pathogen | COVID-19 | 3612 (35.4%) | 2851 (27.1%) |
| | Influenza (pandemic or seasonal) | 10 (0.1%) | 17 (0.2%) |
| | Other | 809 (7.9%) | 902 (8.6%) |
| | None identified | 5778 (56.6%) | 6736 (64.1%) |
| Mortality status | Missing | 9 | 11 |
| | Alive | 8341 (81.8%) | 9132 (87%) |
| | Dead | 1859 (18.2%) | 1363 (13%) |
| | Death with organ support* | 439 (23.6%) | 250 (18.3%) |
| | Death with no organ support* | 1420 (76.4%) | 1113 (81.7%) |
| Organ support | Respiratory | 1165 (11.4%) | 769 (7.3%) |

*(Continued)*

**Table 4.** (Continued)

| Characteristic | Statistic/level | Adult men | Adult women |
|---|---|---|---|
| | Cardiovascular | 360 (3.5%) | 151 (1.4%) |
| | Renal | 155 (1.5%) | 61 (0.6%) |
| | Any | 1241 (12.2%) | 805 (7.7%) |

[†]N = 193 omitted due to missing sex

[*]Denominator = total deaths in category

Table 5 reports the characteristics and outcomes of adults in different ethnic groups. Black or Asian adults tended to be younger than White adults, were less likely to have impaired performance status, be admitted to hospital or die, but were more likely to require organ support or have a positive COVID-19 test. Comorbidities also varied between ethnic groups.

Table 6 shows the characteristics and outcomes of admitted adults with subsequent positive COVID-19 testing and admitted patients with negative or no testing. Age, presenting characteristics, performance status and comorbidities (except chronic lung disease) did not differ markedly between the two groups, but adults with confirmed COVID-19 were more likely to die or require organ support.

## Discussion

Our study describes the presentation of suspected COVID-19 to EDs across the United Kingdom over the first wave of the pandemic. This large, generalizable cohort allows us to characterise the challenge faced by EDs, identify important differences between demographic groups and guide planning for future emergency care.

Adults presenting to the ED with suspected COVID-19 tended to have severe illness, with relatively high NEWS2 scores and abnormal respiratory physiology, and a correspondingly high rate of admission and adverse outcome. Children had a much lower rate of admission and a very low rate of adverse outcome. Adults were also much more likely to have confirmed COVID-19 than children. Suspected COVID-19 in adults and children could therefore be considered as different entities, requiring different approaches to triage, diagnosis and management.

A number of policies were implemented during the pandemic to reduce unnecessary ED attendances with suspected COVID-19. The UK National Health Service advised people with suspected COVID-19 to use the online or telephone NHS111 service rather than attend the ED directly. Some ambulance services avoided transferring people to the ED if they did not have features of severe disease. Our findings suggest that these approaches resulted in an adult ED population with severe illness and high rate of admission. Further research is underway as part of the PRIEST study to determine whether this was achieved at the expense of delayed hospital admission for some cases.

Adults admitted with suspected COVID-19 that was subsequently confirmed were more than twice as likely to die or receive organ support as those who did not have COVID-19 confirmed, despite having similar age, performance status and comorbidities (expect chronic lung disease). Admission with COVID-19 therefore confers a markedly worse prognosis compared to similar presentations. We are only aware of one other study comparing ED presentations in this way—a small single centre study from San Francisco showing no difference in mortality [16].

**Table 5. Characteristics and outcomes of different ethnic groups among adults.**

| Characteristic | Statistic/level | UK/Irish/ other white | Asian | Black/ African/ Caribbean | Mixed/ Multiple groups | Other | Unknown |
|---|---|---|---|---|---|---|---|
| Age (years) | N | 14243 | 1044 | 640 | 247 | 519 | 4215 |
| | Mean (SD) | 64.5 (19.5) | 52.8 (17.8) | 55 (17.7) | 52.8 (19.3) | 51.2 (18.5) | 60.6 (19.7) |
| | Median (IQR) | 67 (51,81) | 50 (40,66) | 54 (41.5,67) | 52 (36,69) | 48 (38,64) | 61 (46,77) |
| Sex | Missing | 129 | 11 | 6 | 4 | 5 | 38 |
| | Male | 6858 (48.6%) | 531 (51.4%) | 309 (48.7%) | 104 (42.8%) | 269 (52.3%) | 2138 (51.2%) |
| | Female | 7256 (51.4%) | 502 (48.6%) | 325 (51.3%) | 139 (57.2%) | 245 (47.7%) | 2039 (48.8%) |
| Presenting features | Cough | 8749 (61.4%) | 717 (68.7%) | 386 (60.3%) | 155 (62.8%) | 342 (65.9%) | 2646 (62.8%) |
| | Shortness of breath | 10662 (74.9%) | 765 (73.3%) | 442 (69.1%) | 178 (72.1%) | 388 (74.8%) | 3151 (74.8%) |
| | Fever | 6756 (47.4%) | 650 (62.3%) | 329 (51.4%) | 127 (51.4%) | 288 (55.5%) | 2132 (50.6%) |
| Symptom duration (days) | N | 12891 | 988 | 601 | 232 | 494 | 3684 |
| | Mean (SD) | 7.6 (8.7) | 9.3 (8.9) | 9.1 (9.5) | 8.8 (8.8) | 8.7 (7.7) | 8.3 (9.5) |
| | Median (IQR) | 5 (2,10) | 7 (3,13) | 7 (3,14) | 7 (3,10.5) | 7 (3,12) | 6 (2,10) |
| Respiratory rate (breaths/ min) | N | 13898 | 1013 | 617 | 239 | 502 | 4094 |
| | Mean (SD) | 23.2 (6.8) | 24.2 (8.2) | 23.7 (7.8) | 22.5 (7.2) | 22.4 (6.6) | 23.3 (7.1) |
| | Median (IQR) | 22 (18,26) | 22 (18,28) | 21 (18,28) | 20 (18,25) | 20 (18,24) | 21 (18,26) |
| Oxygen saturation (%) | N | 14079 | 1031 | 634 | 245 | 513 | 4147 |
| | Mean (SD) | 94.5 (6.9) | 95 (7.6) | 95.3 (7) | 95.6 (5.9) | 95.5 (6.4) | 94.8 (6.4) |
| | Median (IQR) | 96 (94,98) | 97 (95,98) | 97 (95,99) | 97 (95,99) | 97 (95,98) | 96 (94,98) |
| NEWS2 score | N | 14062 | 1021 | 632 | 241 | 509 | 4146 |
| | Mean (SD) | 4.5 (3.3) | 4.2 (3.3) | 4.1 (3.3) | 3.8 (3.3) | 3.7 (3.2) | 4.4 (3.3) |
| | Median (IQR) | 4 (2,7) | 4 (1,6) | 4 (1,6) | 3 (1,6) | 3 (1,6) | 4 (2,7) |
| Comorbidities | Hypertension | 4576 (32.1%) | 338 (32.4%) | 253 (39.5%) | 61 (24.7%) | 105 (20.2%) | 1104 (26.2%) |
| | Heart Disease | 3563 (25%) | 158 (15.1%) | 66 (10.3%) | 28 (11.3%) | 56 (10.8%) | 831 (19.7%) |
| | Diabetes | 2743 (19.3%) | 334 (32%) | 175 (27.3%) | 59 (23.9%) | 67 (12.9%) | 751 (17.8%) |
| | Other chronic lung disease | 2938 (20.6%) | 70 (6.7%) | 45 (7%) | 29 (11.7%) | 47 (9.1%) | 638 (15.1%) |
| | Asthma | 2400 (16.9%) | 160 (15.3%) | 99 (15.5%) | 36 (14.6%) | 63 (12.1%) | 652 (15.5%) |
| | Renal impairment | 1415 (9.9%) | 86 (8.2%) | 63 (9.8%) | 17 (6.9%) | 23 (4.4%) | 330 (7.8%) |
| | Active malignancy | 865 (6.1%) | 26 (2.5%) | 22 (3.4%) | 7 (2.8%) | 12 (2.3%) | 188 (4.5%) |
| | Immunosuppression | 445 (3.1%) | 33 (3.2%) | 29 (4.5%) | 7 (2.8%) | 13 (2.5%) | 104 (2.5%) |
| | Steroid therapy | 414 (2.9%) | 19 (1.8%) | 14 (2.2%) | 4 (1.6%) | 15 (2.9%) | 91 (2.2%) |
| | No Chronic disease | 3452 (24.2%) | 380 (36.4%) | 189 (29.5%) | 97 (39.3%) | 257 (49.5%) | 1423 (33.8%) |
| Performance status | Missing | 706 | 28 | 13 | 6 | 21 | 306 |
| | Unrestricted normal activity | 6549 (48.4%) | 744 (73.2%) | 356 (56.8%) | 180 (74.7%) | 367 (73.7%) | 2345 (60%) |
| | Limited strenuous activity, can do light | 1755 (13%) | 84 (8.3%) | 81 (12.9%) | 22 (9.1%) | 40 (8%) | 391 (10%) |
| | Limited activity, can self care | 2095 (15.5%) | 79 (7.8%) | 70 (11.2%) | 23 (9.5%) | 36 (7.2%) | 478 (12.2%) |
| | Limited self care | 2058 (15.2%) | 50 (4.9%) | 54 (8.6%) | 9 (3.7%) | 32 (6.4%) | 446 (11.4%) |

*(Continued)*

**Table 5.** (Continued)

| Characteristic | Statistic/level | UK/Irish/ other white | Asian | Black/ African/ Caribbean | Mixed/ Multiple groups | Other | Unknown |
|---|---|---|---|---|---|---|---|
| | Bed/chair bound, no self care | 1080 (8%) | 59 (5.8%) | 66 (10.5%) | 7 (2.9%) | 23 (4.6%) | 249 (6.4%) |
| Admitted at initial assessment | Missing | 22 | 1 | 0 | 0 | 0 | 22 |
| | No | 4329 (30.4%) | 445 (42.7%) | 251 (39.2%) | 108 (43.7%) | 262 (50.5%) | 1472 (35.1%) |
| | Yes | 9892 (69.6%) | 598 (57.3%) | 389 (60.8%) | 139 (56.3%) | 257 (49.5%) | 2722 (64.9%) |
| Respiratory pathogen | COVID-19 | 4278 (30%) | 440 (42.1%) | 261 (40.8%) | 68 (27.5%) | 170 (32.8%) | 1304 (30.9%) |
| | Influenza (pandemic or seasonal) | 23 (0.2%) | 1 (0.1%) | 0 (0%) | 0 (0%) | 0 (0%) | 3 (0.1%) |
| | Other | 1361 (9.6%) | 65 (6.2%) | 29 (4.5%) | 16 (6.5%) | 19 (3.7%) | 231 (5.5%) |
| | None identified | 8581 (60.2%) | 538 (51.5%) | 350 (54.7%) | 163 (66%) | 330 (63.6%) | 2677 (63.5%) |
| Mortality status | Missing | 3 | 0 | 0 | 0 | 0 | 17 |
| | Alive | 11903 (83.6%) | 927 (88.8%) | 566 (88.4%) | 221 (89.5%) | 473 (91.1%) | 3552 (84.6%) |
| | Dead | 2337 (16.4%) | 117 (11.2%) | 74 (11.6%) | 26 (10.5%) | 46 (8.9%) | 646 (15.4%) |
| | Death with organ support* | 442 (18.9%) | 40 (34.2%) | 30 (40.5%) | 13 (50%) | 17 (37%) | 151 (23.4%) |
| | Death with no organ support* | 1895 (81.1%) | 77 (65.8%) | 44 (59.5%) | 13 (50%) | 29 (63%) | 495 (76.6%) |
| Organ support | Respiratory | 1189 (8.3%) | 139 (13.3%) | 93 (14.5%) | 31 (12.6%) | 53 (10.2%) | 439 (10.4%) |
| | Cardiovascular | 278 (2%) | 58 (5.6%) | 45 (7%) | 5 (2%) | 14 (2.7%) | 117 (2.8%) |
| | Renal | 115 (0.8%) | 22 (2.1%) | 31 (4.8%) | 3 (1.2%) | 5 (1%) | 42 (1%) |
| | Any | 1264 (8.9%) | 149 (14.3%) | 102 (15.9%) | 34 (13.8%) | 53 (10.2%) | 456 (10.8%) |

*Denominator = total deaths in category

Men and women presented to the ED with suspected COVID-19 in almost equal numbers, but men were more likely to be admitted, have positive COVID-19 testing, receive organ support and die. This may be explained by age and comorbidities. Previous studies have shown a male majority of around 60% among admitted patients [7–10, 17–19]. Petrilli *et al* included patients managed as outpatients or discharged from the ED in their cohort and report similar findings to us, with an equal ratio presenting but men more likely to be admitted [20].

Black or Asian adults tended to be younger than White adults, had less impairment of performance status, and were less likely to be admitted to hospital or die, but were more likely to require organ support or have a positive COVID-19 test. A recent systematic review [21] suggested Black or Asian people are at an increased risk of acquiring COVID-19 and a greater risk of worse clinical outcomes compared to White people. Most studies in the review were from the United States, where social imbalances and inequalities in the access to health care may explain these increased risks. Harrison *et al* studied admitted patients with a high likelihood of COVID-19 infection across UK hospitals over the same time period as our study and showed that higher mortality among the White population was explained by age on multivariable analysis [22]. In contrast, Price-Heywood *et al* found that high mortality associated with Black ethnicity in Louisiana was explained by sociodemographic and clinical characteristics [23], while

**Table 6. Characteristics and outcomes of admitted adult patients with (N = 5768) and without (N = 8229) positive COVID-19 test.**

| Characteristic | Statistic/level | COVID-19 positive | COVID-19 negative or not tested |
|---|---|---|---|
| Age (years) | N | 5768 | 8229 |
| | Mean (SD) | 69.8 (16.6) | 68.4 (17.8) |
| | Median (IQR) | 73 (58,83) | 72 (57,82) |
| Sex | Missing | 53 | 82 |
| | Male | 3282 (57.4%) | 4140 (50.8%) |
| | Female | 2433 (42.6%) | 4007 (49.2%) |
| Presenting features | Cough | 3722 (64.5%) | 4633 (56.3%) |
| | Shortness of breath | 4390 (76.1%) | 6158 (74.8%) |
| | Fever | 3425 (59.4%) | 3629 (44.1%) |
| Symptom duration (days) | N | 5199 | 7278 |
| | Mean (SD) | 6.9 (6.3) | 7 (8.9) |
| | Median (IQR) | 6 (2,10) | 3 (2,8) |
| Respiratory rate (breaths/min) | N | 5634 | 8060 |
| | Mean (SD) | 25.6 (7.8) | 23.9 (6.9) |
| | Median (IQR) | 24 (20,29) | 22 (19,28) |
| Oxygen saturation (%) | N | 5710 | 8152 |
| | Mean (SD) | 92.7 (7.8) | 94.1 (7) |
| | Median (IQR) | 95 (91,97) | 96 (93,98) |
| NEWS2 score | N | 5711 | 8146 |
| | Mean (SD) | 6.1 (3.2) | 5.2 (3.2) |
| | Median (IQR) | 6 (4,8) | 5 (3,7) |
| Comorbidities | Hypertension | 2251 (39%) | 3000 (36.5%) |
| | Heart Disease | 1605 (27.8%) | 2457 (29.9%) |
| | Diabetes | 1591 (27.6%) | 1885 (22.9%) |
| | Other chronic lung disease | 978 (17%) | 2189 (26.6%) |
| | Asthma | 770 (13.3%) | 1276 (15.5%) |
| | Renal impairment | 769 (13.3%) | 959 (11.7%) |
| | Active malignancy | 282 (4.9%) | 693 (8.4%) |
| | Immunosuppression | 181 (3.1%) | 309 (3.8%) |
| | Steroid therapy | 160 (2.8%) | 288 (3.5%) |
| | No Chronic disease | 1158 (20.1%) | 1406 (17.1%) |
| Performance status | Missing | 232 | 504 |
| | Unrestricted normal activity | 2224 (40.2%) | 2989 (38.7%) |
| | Limited strenuous activity, can do light | 605 (10.9%) | 1160 (15%) |
| | Limited activity, can self care | 856 (15.5%) | 1625 (21%) |
| | Limited self care | 1128 (20.4%) | 1286 (16.6%) |
| | Bed/chair bound, no self care | 723 (13.1%) | 665 (8.6%) |
| Mortality status | Missing | 0 | 1 |
| | Alive | 3918 (67.9%) | 6952 (84.5%) |
| | Dead | 1850 (32.1%) | 1276 (15.5%) |
| | Death with organ support* | 471 (25.5%) | 208 (16.3%) |
| | Death with no organ support* | 1379 (74.5%) | 1068 (83.7%) |
| Organ support | Respiratory | 1235 (21.4%) | 661 (8%) |
| | Cardiovascular | 379 (6.6%) | 128 (1.6%) |
| | Renal | 151 (2.6%) | 65 (0.8%) |
| | Any | 1278 (22.2%) | 729 (8.9%) |

*Denominator = total deaths in category

Petrelli *et al* showed that Hispanic ethnicity in New York was associated with an increased risk of hospital admission but not of critical illness [20]. These findings suggest a complex interaction between underlying demographics and comorbidities, susceptibility to COVID-19 and use of health services may explain differences between ethnic groups.

Our study is based on a large and generalizable cohort covering the first wave of the pandemic, but has some limitations. A combination of prospective and retrospective data collection was used, and infection control measures limited our ability to collect data directly from patients. Reliance on clinical records may have underestimated the prevalence of some presenting features and co-morbidities, and resulted in missing data for some variables. Selection of cases was based on subjective clinical judgement that COVID-19 was a suspected diagnosis, which may have been applied in a variable manner between clinicians and between sites. Our analysis was limited to describing the cohort rather than using multivariable analysis to explain the observed differences between groups. We felt that the latter analysis would need to be based on a clear theoretical rationale and inclusion of appropriate covariates, which would be beyond the scope of this study. Finally, the use of our data to guide planning of emergency care may be limited by changes in the characteristics of patients presenting in future waves of the pandemic. Further research is therefore required to determine the characteristics of patients in future waves.

## Conclusion

We have shown important differences between patient groups presenting to the ED with suspected COVID-19. Adults and children differ markedly and require different approaches to emergency triage. Admission and adverse outcome rates among adults suggest that policies to avoid unnecessary ED attendance achieved their aim. Subsequent COVID-19 confirmation confers a worse prognosis and greater need for organ support.

## Supporting information

**S1 Appendix. Standardised data collection form.**
(PDF)

**S2 Appendix. Follow-up form.**
(PDF)

**S3 Appendix. Study steering committee.**
(DOCX)

**S4 Appendix. Site research staff.**
(DOCX)

**S5 Appendix. Supporting research staff.**
(DOCX)

## Acknowledgments

We thank Katie Ridsdale for clerical assistance with the study, Erica Wallis (Sponsor representative), Matt Burnsall and Mike Bradburn for additional statistical support, all members of the Study Steering Committee (SDF_ S3 Appendix: Study Steering Committee) and the site research staff who delivered the data for the study (SF_S4 Appendix: Site Research Staff), and the research team at the University of Sheffield past and present (SF_S5 Appendix: Supporting Research Staff).

## Author Contributions

**Conceptualization:** Steve Goodacre, Andrew Bentley, Kirsty Challen, Chris Fitzsimmons, Tim Harris, Fiona Lecky, Andrew Lee, Ian Maconochie, Darren Walter.

**Data curation:** Ben Thomas, Amanda Loban, Simon Waterhouse, Richard Simmonds, Katie Biggs, Jose Schutter, Sarah Connelly, Elena Sheldon, Jamie Hall, Emma Young.

**Formal analysis:** Steve Goodacre, Ellen Lee, Laura Sutton.

**Funding acquisition:** Steve Goodacre, Katie Biggs, Andrew Bentley, Kirsty Challen, Chris Fitzsimmons, Tim Harris, Fiona Lecky, Andrew Lee, Ian Maconochie, Darren Walter.

**Investigation:** Ben Thomas, Amanda Loban, Simon Waterhouse, Richard Simmonds, Katie Biggs, Carl Marincowitz, Jose Schutter, Sarah Connelly, Elena Sheldon, Jamie Hall, Emma Young.

**Methodology:** Steve Goodacre, Ellen Lee, Laura Sutton, Andrew Bentley, Kirsty Challen, Chris Fitzsimmons, Tim Harris, Fiona Lecky, Andrew Lee, Ian Maconochie, Darren Walter.

**Project administration:** Ben Thomas, Katie Biggs.

**Supervision:** Steve Goodacre.

**Validation:** Ben Thomas, Ellen Lee, Laura Sutton, Katie Biggs, Carl Marincowitz.

**Writing – original draft:** Steve Goodacre, Ben Thomas, Ellen Lee, Laura Sutton, Katie Biggs.

**Writing – review & editing:** Ben Thomas, Ellen Lee, Laura Sutton, Amanda Loban, Simon Waterhouse, Richard Simmonds, Katie Biggs, Carl Marincowitz, Jose Schutter, Sarah Connelly, Elena Sheldon, Jamie Hall, Emma Young, Andrew Bentley, Kirsty Challen, Chris Fitzsimmons, Tim Harris, Fiona Lecky, Andrew Lee, Ian Maconochie, Darren Walter.

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
