## [Decision Letter · Decision Letter 0]

15 Sep 2020

PONE-D-20-25198

Characterisation of 22446 patients attending UK emergency departments with suspected COVID-19 infection: Observational cohort study

PLOS ONE

Dear Dr. Goodacre,

Thank you for submitting your manuscript to PLOS ONE. After careful consideration, we feel that it has merit but does not fully meet PLOS ONE’s publication criteria as it currently stands. Therefore, we invite you to submit a revised version of the manuscript that addresses the points raised during the review process.

We look forward to receiving your revised manuscript.

Kind regards,

Walter R. Taylor

Academic Editor

PLOS ONE

Journal Requirements:

2.We note that you have indicated that data from this study are available upon request. PLOS only allows data to be available upon request if there are legal or ethical restrictions on sharing data publicly. For information on unacceptable data access restrictions, please see http://journals.plos.org/plosone/s/data-availability#loc-unacceptable-data-access-restrictions.

3.Thank you for stating the following in the Competing Interests section:

[All authors declare grant funding to their employing institutions from the National Institute for Health Research, as outlined under financial disclosure information.

SG is Deputy Director of the NIHR Health Technology Assessment (HTA) Programme, which funded the study, and chairs the NIHR HTA commissioning commitee.].

4. One of the noted authors is a group or consortium [The PRIEST Research Group]. In addition to naming the author group, please list the individual authors and affiliations within this group in the acknowledgments section of your manuscript. Please also indicate clearly a lead author for this group along with a contact email address.

5. Please respond by return e-mail with an updated version of your manuscript to amend either the abstract on the online submission form or the abstract in the manuscript so that they are identical. We can make any changes on your behalf.

6. Please include captions for your Supporting Information files at the end of your manuscript, and update any in-text citations to match accordingly. Please see our Supporting Information guidelines for more

Additional Editor Comments (if provided):

Dear Dr. Goodacre,

I have received comments from one reviewer.

This reviewer has raised some interesting points and I look forward to seeing your responses.

yours sincerely,

Walter Taylor.

Reviewers' comments:

Reviewer's Responses to Questions

**Comments to the Author**

1. Is the manuscript technically sound, and do the data support the conclusions?

Reviewer #1: Yes

2. Has the statistical analysis been performed appropriately and rigorously? 

Reviewer #1: Yes

3. Have the authors made all data underlying the findings in their manuscript fully available?

Reviewer #1: Yes

4. Is the manuscript presented in an intelligible fashion and written in standard English?

Reviewer #1: Yes

5. Review Comments to the Author

Reviewer #1: The manuscript presented by Goodacre and colleagues adds important epidemiological information to the understanding of the SARS-CoV-2 epidemic in the UK. The authors analysed the clinical data of 22446 patients admitted with suspected COVID-19 to 70 emergency rooms in the UK from March 26th until May 28th 2020. In summery

- Male sex as a risk factor for a more severe cause of COVID-19

- COVID-19 results in a more severe course than other respiratory diseases even when the groups have similar rates of comorbidities in the beginning.

- And to some interest, ethnical differences which (to my knowledge) can’t be explained by biological facts. Black and Asian adults were roughly 15 years younger, had a better performance status, were less likely to be admitted to hospital and were less likely to die. Nevertheless, they had a higher rate of COVID-19 positive tests and needed more organ support.

For me the last point is of importance. The authors state that Black and Asian patients might have a higher risk for a more severe COVID-19 course. But the review article they cite (Ref. 21) may not be optimal to prove this claim. Most studies included in this review reporting differences in outcome depending on ethnicity were from the US. Due to fundamental differences in the US and UK health care systems, the result of these studies are rather a surrogate for social imbalances and inequalities in the access to health care than a prove for biological differences.

The study is well executed and the presentation of the results are fine. However, in the discussion the authors claim that their study allows to “…guide planning for future emergency care.” I would like to question this in at least in part. Because the manuscript describes the first wave. Presumable some of the parameters will be the same during the ongoing pandemic but others will change; the introduction of the virus will happen through other routes, precaution measures for populations at risk are still in place and pandemic response will be based on the experience of the past. How an epidemic may change you see for example in Germany, the average age of patients diagnosed with CODIV-19 dropped from 52 years in May 2020 to 32 years in August (national surveillance data from the Robert Koch Institute). Hospital and ICU admission rates dropped from ≈ 10% and 4% to 3% and >1% respectively.

6. PLOS authors have the option to publish the peer review history of their article (what does this mean?). If published, this will include your full peer review and any attached files.

Reviewer #1: No

---

## [Author Response · Author response to Decision Letter 0]

20 Sep 2020

Thank you for considering our paper and for the reviewer’s thoughtful comments. We have revised our paper to address these comments and we provide our responses below. The amendments are marked using track changes in the revised manuscript. We have also made a very small amendment to the numbers in the manuscript as a result of identifying a duplicated case. The reduction from 22446 to 22445 cases resulted in no significant change to the reported findings.

The authors state that Black and Asian patients might have a higher risk for a more severe COVID-19 course. But the review article they cite (Ref. 21) may not be optimal to prove this claim. Most studies included in this review reporting differences in outcome depending on ethnicity were from the US. Due to fundamental differences in the US and UK health care systems, the result of these studies are rather a surrogate for social imbalances and inequalities in the access to health care than a proof for biological differences.

Thank you for highlighting this. We have added a sentence to the discussion to make this point. We have also added a reference for the Harrison study, which was undertaken in the UK.

The study is well executed and the presentation of the results are fine. However, in the discussion the authors claim that their study allows to “…guide planning for future emergency care.” I would like to question this in at least in part. Because the manuscript describes the first wave. Presumable some of the parameters will be the same during the ongoing pandemic but others will change; the introduction of the virus will happen through other routes, precaution measures for populations at risk are still in place and pandemic response will be based on the experience of the past. How an epidemic may change you see for example in Germany, the average age of patients diagnosed with CODIV-19 dropped from 52 years in May 2020 to 32 years in August (national surveillance data from the Robert Koch Institute). Hospital and ICU admission rates dropped from ≈ 10% and 4% to 3% and >1% respectively.

Thank you for highlighting this issue. We have added a couple of sentences to the discussion to acknowledge this limitation and identify the need for further research to characterise patients presenting in future waves of the pandemic.

---

## [Editor Report · Decision Letter 1]

23 Sep 2020

Characterisation of 22445 patients attending UK emergency departments with suspected COVID-19 infection: Observational cohort study

PONE-D-20-25198R1

Dear Dr. Goodacre,,

We’re pleased to inform you that your manuscript has been judged scientifically suitable for publication and will be formally accepted for publication once it meets all outstanding technical requirements.

Kind regards,

Walter R. Taylor

Academic Editor

PLOS ONE

Additional Editor Comments (optional):

None
---

## [Editor Report · Acceptance letter]

17 Nov 2020

PONE-D-20-25198R1 

Characterisation of 22445 patients attending UK emergency departments with suspected COVID-19 infection: Observational cohort study 

Dear Dr. Goodacre:

I'm pleased to inform you that your manuscript has been deemed suitable for publication in PLOS ONE. Congratulations! Your manuscript is now with our production department. 

Kind regards, 

on behalf of

Dr. Walter R. Taylor 

Academic Editor

PLOS ONE